# RAG Knowledge Online Correction with Conversation-Based User Feedback

## Abstract

Retrieval-Augmented Generation (RAG) is a promising way to enhance LLMs by integrating external knowledge. However, its performance degrades when the knowledge contains errors. When users encounter such errors, the typical correction process involves users reporting the errors, after which server providers investigate the knowledge base to identify and fix the issues. This process is often time-consuming, and in the meantime, other users continue to encounter the same errors, leading to poor user experience. To address this challenge, we propose a new task, **Knowledge Online Correction**, which focuses on correcting errors immediately after they are pointed out by users through conversation-based feedback. To evaluate this task, we conducted a preliminary user study and developed a new benchmark, **ConvCorrect**. To address this task, we propose a Multi-step Knowledge Online Correction method(**MT-KOC**), an online knowledge correction method that automatically corrects errors in real time based on a dynamic action search algorithm. Empirical results shows that MT-KOC outperforms baseline methods, achieving higher accuracy in the knowledge online correction task.

## 1 Introduction

Large Language Models (LLMs) have achieved remarkable success in real world applications, but still face challenges such as hallucination and outdated knowledge (Huang et al., 2025; Zhang et al., 2023). Retrieval-Augmented Generation (RAG) has emerged as a promising solution by integrating external knowledge to mitigate these issues (Lewis et al., 2020; Gao et al., 2023b). However, its performance degrades when the underlying knowledge base contains errors (e.g., Fig. 1A). When users encounter such errors, the typical correction process involves tedious manual intervention: users report the issue, and server providers investigate the knowledge base to identify and correct the errors (Fig. 1B). This process is often time-consuming, potentially taking several hours to multiple working days (McGraw, 2025). During this period, the knowledge errors persist and continue to mislead users, resulting in a poor overall experience. To address the issue of inefficiency, some re-

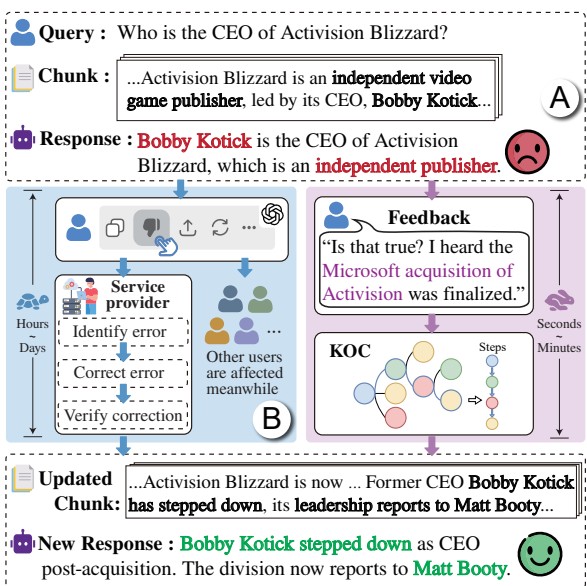

Figure 1: Comparison between slow manual correction (left) and rapid knowledge online correction (right).

cent studies have proposed to automatically correct the errors in real time through internet retrieval (e.g., Wikipedia) (Yan et al., 2024). However, the effectiveness of these methods depends on the accuracy of the internet retrieval results. If the internet retrieval results are incorrect, these methods have no alternative mechanism to resolve the errors.

To tackle this issue, we drew inspiration from the **Formative Assessment with Feedback** model in education (McKenzie et al., 2017). In this model, teachers provide real-time, conversation-based feedback during discussions, enabling students to quickly identify and correct misconceptions and thereby accelerate learning. Motivated by this model, we propose a new strategy, where users act as teachers, offering conversation-based feedback when they identify errors. For example, users can indicate whether an error exists and suggest how it should be corrected. Such feedback can serve as additional guidance for error correction, helping to overcome the limitations of existing methods that rely solely on internet retrieval results. Based on this new strategy, we define a new task, Knowledge Online Correction (KOC), which leverages conversation-based user feedback to correct knowledge errors in real time.

While this new strategy has the potential to help correct knowledge errors in real time, how to effectively leverage user feedback is still unclear, presenting two major challenges. **Challenge 1**: There are no established benchmarks for this task. History has demonstrated that high-quality benchmarks (e.g., ImageNet (Deng et al., 2009), GLUE (Wang et al., 2019)) can effectively advance the development of models for specific tasks. However, developing benchmarks for the KOC task is non-trivial due to the diverse and open-ended nature of conversation-based feedback. Therefore, how to ensure that the benchmark is both comprehensive and representative remains an open question. **Challenge 2**: It is unclear how to effectively leverage the conversation-based user feedback for error correction. Conversation-based feedback can take many forms, from precise, well-structured corrections to vague hints that an error exists. Effectively distinguishing useful feedback from noisy or misleading input is critical, as is determining how to integrate this feedback into model updates in a way that effectively improves performance.

In this paper, we first developed a new benchmark, ConvCorrect (**Challenge 1**). To construct this benchmark, we first conducted a preliminary user study. From this preliminary study, we summarized five error types in the knowledge bases and how users provide feedback for these errors. Based on this summarization, we synthesize errors in the knowledge bases and the conversation-based feedback, which was further verified by humans to ensure quality. Based on this benchmark, we developed MT-KOC for the KOC task (**Challenge 2**). MT-KOC uses a dynamic action search algorithm to identify an optimal sequence of edits for progressive correction, ensuring better alignment with conversation-based feedback. The experiments show that our method outperforms the existing SOTA methods by 5.88% on average in terms of F1-score.

In summary, our contributions are threefold:

1. We develop ConvCorrect, the first benchmark designed and supported by a preliminary user study to evaluate knowledge online correction methods.

2. We propose MT-KOC, a method based on a dynamic action search algorithm to address the challenges of online knowledge base correction.

3. We validate our approach through empirical experiments, demonstrating the effectiveness of MT-KOC, and through a user study that confirms the rational design of our benchmark.

## 2 RELATED WORK

### 2.1 RETRIEVAL-AUGMENTED GENERATION

Retrieval-Augmented Generation (RAG) has been established as an effective paradigm for mitigating hallucinations in Large Language Models (LLMs) (Brown et al., 2020; Chowdhery et al., 2022; Touvron et al., 2023) by grounding them in external knowledge (Lewis et al., 2020; Gao et al., 2023b; Guu et al., 2020; Izacard & Grave, 2021). Extensive research has focused on optimizing the RAG pipeline, with significant advancements in query formulation (Zheng et al., 2024; Ma et al., 2023), dynamic and adaptive retrieval strategies (Asai et al., 2024; Jeong et al., 2024), and post-retrieval context refinement (Yu et al., 2024; Li et al., 2025; Xu et al., 2024).

However, a majority of these methods operate under the assumption that the knowledge source itself is accurate. Their objective is to better *utilize* existing knowledge, not to *correct* the knowledge base. While a parallel line of research explores editing the model's internal, parametric knowledge (Meng et al., 2023; Zhang et al., 2024), this is distinct from correcting external, non-parametric knowledge

sources. Advanced frameworks (Yan et al., 2024; Wang et al., 2025) focus on rectifying the answer for a single turn, rather than propagating the fix back to the underlying source. To address this critical gap, we introduce the novel task of **Knowledge Online Correction (KOC)**, which aims to rectify the knowledge source in real-time, driven by user feedback.

## 2.2 MULTI-AGENT SYSTEMS

Multi-Agent Systems (MAS), wherein complex tasks are decomposed and solved by multiple collaborative agents, have demonstrated significant efficacy and have been widely explored (Wang et al., 2024; Xi et al., 2025). Recent advancements in MAS have progressed along two main fronts: developing increasingly sophisticated collaborative frameworks, evolving from structured workflows to large-scale engineering platforms (Hong et al., 2024; Qian et al., 2024), and enhancing the individual agent's capabilities with mechanisms for complex reasoning, reflection, and self-correction (Yao et al., 2023b; Shinn et al., 2023; Gou et al., 2024; Madaan et al., 2023).

Our approach is inspired by the core MAS principle of solving problems via a sequence of planned actions. However, our work introduces key innovations. Most MAS are **generative**, tasked with creating content like code or reports. While some agents exhibit self-correction, this is typically aimed at refining their own generated output. We, in contrast, use this paradigm for a corrective task—our KOC, which targets an external knowledge source. Furthermore, our system is designed to be **feedback-driven**, specifically for the purpose of dynamic knowledge correction in RAG systems.

## 3 BENCHMARK

### 3.1 PROBLEM FORMULATION

The KOC task is formally defined as follows. Given a user query $Q$, a knowledge chunk $K$ is retrieved first, which includes errors. Based on the chunk $K$, an answer $A$ is generated by an LLM: $A = LLM(Q, K)$. After receiving the answer, the user responds with conversation-based feedback $F$. Then, the goal of the KOC task is to find a transformation function $T$ such that the answer generated based on the correct version $K_{oracle}$ is similar to the answer generated based on the chunk transformed by $T$:

$$T^* = \underset{T}{\arg\min} \quad \text{Diff}\{\text{LLM}(Q, K^*), \text{LLM}(Q, K_{\text{oracle}})\}$$
$$\text{s.t.} \quad K^* = T(K, F) \tag{1}$$

$\text{Diff}\{\cdot, \cdot\}$ measures the difference between two answers generated by the LLM.

### 3.2 PRELIMINARY USER STUDY

A key challenge in constructing the benchmarks for the KOC task lies in ensuring their comprehensiveness and representativeness. To achieve this, it is essential to understand how users typically provide conversation-based feedback when they encounter errors in LLM-generated answers. Therefore, we conducted a preliminary user study. Based on this study, we identified five common types of errors in LLM answers and further summarized typical conversation-based feedback corresponding to each error type.

**Study setup.** We recruited 12 graduate students majoring in Computer Science for the user study, aged from 21 to 29 years (mean = 24.25, SD = 2.60). All of them have more than one year of experience in developing or using RAG systems. Upon completion, each participant received a $10 compensation, independent of their performance.

**Datasets.** The study was constructed on 1,200 samples randomly selected from the Neural-Bridge Dataset (Neural Bridge AI, 2024b;a). Each sample contains a query Q and a correct knowledge chunk $K_{oracle}$. Then, the chunk of each sample was perturbed to contain errors. To ensure the errors encompass most real-world scenarios, the perturbations are capable of generating any type of error. In the field of databases, updating, adding, and removing are used to cover all types of modifications. Motivated by this, we also include three types of perturbations Silberschatz et al. (2002). (1) *Update:* A key entity (e.g., date, name, location) in the relevant section is updated to

an incorrect but plausible entity of the same type. (2) _Add:_ A contextually coherent but factually incorrect or misleading sentence is added after the sentence containing the relevant information. (3) _Delete:_ A critical phrase or sentence required to answer the query is removed from the relevant section. The chunk of each sample was randomly applied one of the three types of perturbations, resulting in an erroneous knowledge chunk ($K$).F Based on $K$, an answer ($A$) was generated by an LLM.

**Procedure.** The 1,200 samples were randomly distributed to the 12 participants, with each participant receiving 100 samples. For each sample, the participants were asked to assume the role of a user who had received an erroneous answer ($A$) from a conversational AI system after posing a query. Their task was to write a single conversation-based feedback message ($F$) in response to the erroneous answer, reflecting what they would realistically provide in such a scenario.

**Error type.** After the user study, two researchers from our team were tasked with classifying the error type of the answer ($A$) in each of the 1200 samples. Each annotator was responsible for half of the data, and a cross-review was conducted upon completion to ensure consistency. This process resulted in the classification of all samples into five primary error types: **Fully Incorrect**, **Partially Incorrect**, **Fully Missing**, **Partially Missing**, and **Mixed**. For illustrative examples of each error type, please refer to Appendix A.

**Typical conversation-based feedback styles.** After the user study was completed, the same two expert annotators analyzed the feedback ($F$) provided by the participants. Following a similar process of independent annotation and cross-review, they categorized the collected feedback into seven distinct styles. Our analysis revealed that different error types elicited different distributions of feedback styles, as summarized in Table 1.

| Error type | Associated feedback styles |
|---|---|
| Fully incorrect | Direct correction, Error indication |
| Partially incorrect | Direct correction, Error indication, Error localization |
| Fully missing | Direct completion, Missing indication |
| Partially missing | Direct completion, Missing indication |
| Mixed | Direct correction and completion, Error and missing indication |

Table 1: Mapping from Error Types to Feedback Styles derived from the user study.

A key finding is that the relationship between error types and feedback styles is a complex one-to-many mapping. This discovery highlights a fundamental challenge: a simple, low-level textual perturbation can manifest as a wide spectrum of high-level, user-perceived semantic errors. This non-trivial mapping proves that a simplistic approach—mechanically generating feedback based on the perturbation type—would be unrealistic and fail to capture the task's true complexity. Therefore, this empirically derived taxonomy not only reveals the task's difficulty but also serves as the principled blueprint for our benchmark construction.

### 3.3 BENCHMARK CONSTRUCTION

Based on the error types and typical conversation-based feedback styles identified in the preliminary user study, we developed the first benchmark for the KOC task, ConvCorrect.

**Data Source and Pre-processing.** We construct ConvCorrect upon two diverse RAG benchmarks: SQuAD (Rajpurkar et al., 2016) and Neural-Bridge (Neural Bridge AI, 2024b;a), resulting in two splits: ConvCorrect-SQ and ConvCorrect-NB. We first pre-processed these source datasets to create a clean base for perturbation. For SQuAD, which features multiple questions per context, we retained only one unique sample per context, merging the training and test sets to yield 20,958 initial samples. For Neural-Bridge, we combined its two versions (RAG Dataset 12000 & RAG Dataset 1200) and de-duplicated them to obtain 12,893 unique samples. Finally, both datasets underwent sensitive content filtering. This resulted in a final set of 19,903 pristine samples from SQuAD and 12,000 from Neural-Bridge, which formed the basis for our benchmark construction.

**Construction pipeline.** Based on the pre-processed datasets, we constructed ConvCorrect in two main steps: **error generation** and **feedback generation**.

- **Error generation.** Since the knowledge chunks in the source datasets contain no errors, we first perturbed the chunks to contain errors. Similar to the preliminary user study, we consider three types of perturbations here. For each chunk, all three types of perturbations were applied using LLMs (Ding et al., 2024), resulting in three distinct perturbed versions. These perturbed chunks were then classified into the five error types identified in the preliminary user study, also using LLMs. Finally, both the perturbed chunks and their corresponding error classifications were verified by two human evaluators.

- **Feedback generation.** Finally, to comprehensively model the diversity of user responses, we exhaustively generate all applicable feedback styles for *each* classified error. An LLM acting as a **User Simulator** takes the error type label from Stage 2 and, based on the one-to-many mapping from our user study (Table 1), generates multiple feedback messages ($F$) in different styles. For instance, for an error classified as Partially incorrect, the system generates separate samples with Direct Correction, Error Indication, and Error Localization feedback styles.

**Statistics.** The three-stage simulation pipeline, particularly the exhaustive generation strategy in Stage 1 and Stage 3, culminates in the final ConvCorrect benchmark, comprising **124,531** samples for ConvCorrect-SQ and **80,772** for ConvCorrect-NB. This strategy ensures our benchmark has high coverage and diversity in its error-feedback pairings. The detailed statistics of the Benchmark's scale and type distribution are presented in Appendix Table 5.

## 4 METHOD

### 4.1 FRAMEWORK OVERVIEW

To tackle the KOC task, we introduce the **MulTi**-step KOC (MT-KOC) framework. The design of MT-KOC is inspired by the typical correction process followed by humans. Generally, humans first retrieve relevant information from reliable sources, then use this information to correct specific chunks, and finally verify the accuracy of the corrected chunks. Since errors may be diverse and cannot always be resolved in a single step in practice, this process may be repeated multiple times. Therefore, we design a multi-agent system within MT-KOC to address the KOC task through multiple steps. Leveraging this system, we propose a dynamic action search algorithm that identifies the optimal sequence of corrective actions to accurately correct the chunks.

### 4.2 MULTI-AGENT SYSTEM

The multi-agent system consists of four collaborative agents: a **Knowledge Distillation Agent**, an **Action Recommendation Agent**, a **Correction Agent**, and a **Reward Evaluation Agent**.

**Knowledge Distillation Agent.** The correction of knowledge chunks relies on retrieval results, such as those from the Internet. Therefore, the agent retrieves relevant information for the chunks to be corrected. It takes a user query ($Q$) and the initial Knowledge Chunk ($K_0 = K$) as input to extract information ($I$). This process is represented by the function $f_{\text{distill}}$:

$$I = f_{\text{distill}}(Q, K_0) \qquad (2)$$

The function $f_{\text{distill}}$ is realized by the agent, which is an LLM guided by the prompt detailed in Appendix B.1. It is important to note that this agent is not limited to retrieving information from the Internet; it can also retrieve from LLMs, also known as knowledge distillation.

**Action Recommendation Agent.** As pointed out in a recent survey (Tran et al., 2025), the performance of LLMs would be improved if a complex action is decomposed into several sub-actions. Therefore, we also decompose the correction action into three commonly used sub-actions: ADD, DELETE, and REVISE, which are called actions. ADD: Complete missing information in Knowledge Chunk, where the agent specifies the exact information and insertion position. DELETE: Remove

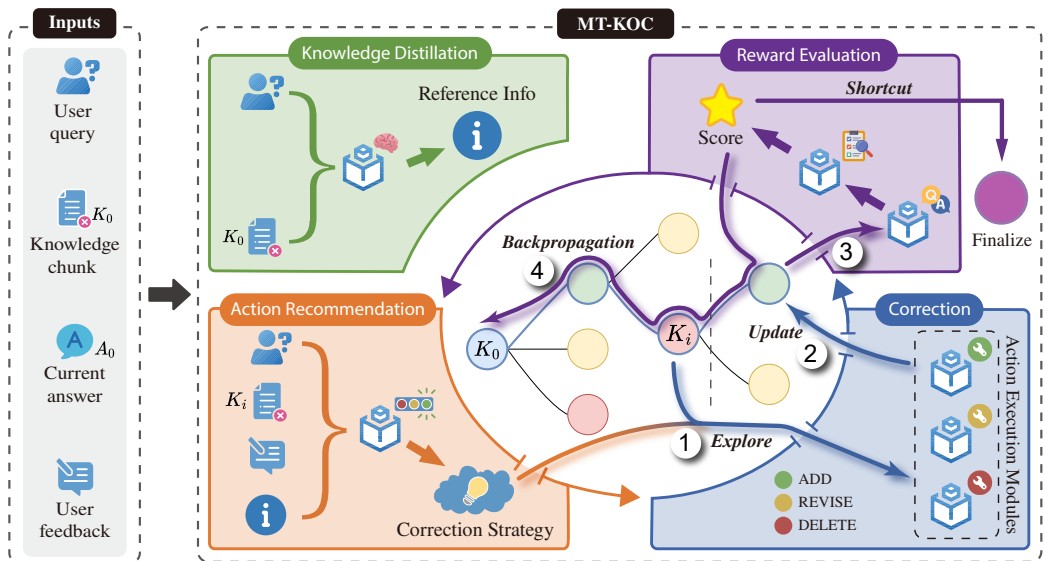

Figure 2: Overview of the MT-KOC framework, illustrating the collaboration across four key agents. The process iteratively explores a search space of possible edits to find an optimal correction path for the Knowledge Chunk, guided by user feedback and a reward-driven mechanism.

redundant or misleading information in the Knowledge Chunk that could interfere with generating the correct answer, with the agent explicitly identifying the target. REVISE: Modify inaccurate or poorly phrased information in the Knowledge Chunk, where the agent provides a revised version. Accordingly, the correction can be represented by a sequence of actions $\{C_1, ..., C_k\}$. Therefore, action recommendation agents recommend suitable actions given the user query ($Q$), current knowledge chunk ($K$), user feedback ($F$), and reference information ($I$). The agent's recommendation logic is implemented using an LLM, its specific prompt is provided in Appendix B.2.

**Correction Agent.** This agent acts as the executor. It is a composite of three specialized modules corresponding to the action types (ADD, DELETE, REVISE). When a single correction action $C_i$ is selected by the Recommendation Agent, this agent dispatches it to the appropriate module for execution, producing the next Knowledge Chunk state, $K_{i+1}$:

$$K_{i+1} = f_{\text{correct}}(K_i, C_i) \tag{3}$$

The function $f_{\text{correct}}$ represents this execution, each module within this agent is also guided by an LLM prompt, as detailed in Appendix B.3.

**Reward Evaluation Agent.** This agent evaluate the corrections based on the conversation-based user feedback ($F$) and reference information ($I$) (Zheng et al., 2023). It takes the new Knowledge Chunk state $K_{i+1}$ and generates the answer $A_{i+1}$ by using LLMs. It then assesses this answer based on the query ($Q$), feedback ($F$), and reference information ($I$) to produce a numerical score, $S_{i+1}$:

$$S_{i+1} = f_{\text{eval}}(K_{i+1}, Q, F, I) \tag{4}$$

The evaluation function $f_{\text{eval}}$ is implemented by the agent, which is prompted to act as an impartial judge. The full prompt, which specifies the scoring criteria and scale, can be found in Appendix B.4.

### 4.3 DYNAMIC ACTION SEARCH

Based on the multi-agent system, it is required to identify the optimal sequence of corrective actions to accurately correct the chunks. Therefore, the transformation $T$ is realized by finding an optimal sequence of discrete correction actions, which we denote as $\mathbf{C}^*$. Let $\text{Apply}(K_0, \mathbf{C})$ be the function that returns the final Knowledge Chunk after applying the entire sequence $\mathbf{C}$ to the initial chunk $K_0$. The optimal sequence $\mathbf{C}^*$ is the one that maximizes the expected reward of this final chunk:

$$\mathbf{C}^* = \arg\max_{\mathbf{C}} \quad \mathbb{E}\left[f_{\text{eval}}\left(\text{Apply}(K_0, \mathbf{C})\right)\right] \tag{5}$$

**1. Initialization.** The process begins with the initial erroneous Knowledge Chunk, $K_0$, which forms the root node of the search tree. The Knowledge Distillation Agent is invoked once to generate a global Reference Information ($I$). Subsequently, the Action Recommendation Agent generates the first set of candidate corrections from this root node.

**2. Iterative Search.** For a predefined number of Search Epochs, the framework executes a search loop consisting of the following steps:

*Exploration:* In each iteration, the Action Recommendation Agent recommends a set of candidate actions, $\mathcal{C}(K_i)$, based on the current Knowledge Chunk state, represented by the current node $K_i$ in the search tree. An action is then selected from the children using the Upper Confidence Bound applied to Trees (UCT) policy (Kocsis & Szepesvári, 2006; Yao et al., 2023a):

$$C_{i+1} = \arg\max_{C \in \mathcal{C}(K_i)} \left( \frac{R(C)}{N(C)} + c\sqrt{\frac{\ln N(K_i)}{N(C)}} \right) \tag{6}$$

Here, $\arg\max\limits_{C \in \mathcal{C}(K_i)}$ selects the action $C_{i+1}$ from the candidate set $\mathcal{C}(K_i)$ that maximizes the UCT score. The UCT score balances exploitation and exploration: the term $\frac{R(C)}{N(C)}$ represents the average reward of action $C$, encouraging the selection of actions with historically high rewards, while $c\sqrt{\frac{\ln N(K_i)}{N(C)}}$ promotes exploration of less-visited actions, where $R(C)$ is the total reward accumulated for action $C$, $N(C)$ is the number of times action $C$ has been selected, $N(K_i)$ is the number of visits to the parent node $K_i$, and $c$ is an exploration constant.

*Updating:* Once an action $C_i$ is selected, the Correction Agent executes it to produce the next Knowledge Chunk state, $K_{i+1}$, which corresponds to a new node in the search tree.

*Evaluation:* The Reward Evaluation Agent then evaluates this new state to obtain a score $S_{i+1}$.

*Backpropagation:* This score is backpropagated up the search tree, updating the reward and visit counts for all actions along the path from the current node back to the root. Specifically, for each action $C$ on the path: $R(C) \leftarrow R(C) + S_{i+1}$ and $N(C) \leftarrow N(C) + 1$.

**3. Final Selection.** The search process terminates, and an optimal corrected Knowledge Chunk ($K^*$) is selected based on one of two conditions:

*Shortcut Mechanism:* If at any point during the iterative search the Reward Evaluation Agent outputs a maximum score, the search is immediately terminated. The Knowledge Chunk corresponding to this high-reward path, represented by the corresponding node in the search tree, is directly returned as the optimal solution ($K^*$).

*Default Selection:* If the Shortcut is not triggered after all Search Epochs are completed, the system then determines the optimal correction sequence $\mathbf{C}^* = (C_0^*, C_1^*, ..., C_L^*)$. This sequence is constructed by starting from the root node ($K_0^* = K_0$) and iteratively selecting the correction with the highest average reward. At each step 'i', the selection of the next correction $C_{i+1}^*$ is made from the children of the current optimal state $K_i^*$:

$$C_{i+1}^* = \arg\max_{C \in \mathcal{C}(K_i^*)} \left( \frac{R(C)}{N(C)} \right) \tag{7}$$

Here, $\arg\max\limits_{C \in \mathcal{C}(K_i^*)}$ selects the action $C_{i+1}^*$ with the highest average reward, ensuring a greedy selection of the best-performing actions to construct the optimal sequence. The final corrected Knowledge Chunk ($K^*$) is then obtained by applying this sequence $\mathbf{C}^*$ to $K_0$.

## 5 EXPERIMENTS

### 5.1 EXPERIMENTAL SETUP

**Datasets.** We evaluate our method on **ConvCorrect**, a benchmark we developed for interactive knowledge correction, consisting of two splits: ConvCorrect-SQ and ConvCorrect-NB (see Section 3

| | ConvCorrect-SQ | | | ConvCorrect-NB | | |
|---|---|---|---|---|---|---|
| **Method** | **F1** | **Precision** | **Recall** | **F1** | **Precision** | **Recall** |
| Perturbed | 5.5 | 3.6 | 11.4 | 36.0 | 35.0 | 37.0 |
| CRAG | 23.1 | 13.8 | 71.6 | 52.0 | 50.3 | 53.8 |
| RARR | 23.0 | 13.8 | 68.3 | 52.2 | 50.4 | 54.2 |
| Astute | 23.5 | 14.0 | 72.8 | 53.0 | 50.2 | 56.2 |
| **MT-KOC** | **24.8** | **14.9** | **73.6** | **56.3** | **54.1** | **58.7** |
| Oracle | 30.3 | 18.2 | 89.8 | 64.7 | 61.3 | 68.6 |

Table 2: Aggregated F1-Scores, Precision, and Recall on the ConvCorrect-SQ and ConvCorrect-NB test sets. Comparison methods are framed by the performance bounds.

for details). Each sample includes the original knowledge chunk ($K_{oracle}$), a perturbed version ($K_0$), the query ($Q$), the erroneous answer ($A_0$), and user feedback ($F$).

**Comparison Methods.** To situate the performance of MT-KOC, we compare it against a diverse set of baselines, along with theoretical upper and lower bounds.

- **Lower Bound (Perturbed)** uses the initial, erroneous knowledge chunk ($K_0$) directly, establishing a performance floor.
- **Upper Bound (Oracle)** uses the ground-truth correct knowledge chunk ($K_{oracle}$), representing the theoretical performance ceiling.
- **CRAG** (Yan et al., 2024) employs a **self-correction** paradigm, using a lightweight retrieval evaluator to judge the necessity of refinement.
- **RARR** (Gao et al., 2023a) utilizes a **multi-step reasoning** approach, prompting the LLM to explicitly "research and revise" its outputs in a verifiable manner.
- **Astute** (Wang et al., 2025) tackles the problem via **knowledge conflict resolution**, identifying and resolving discrepancies between parametric and retrieved knowledge.

**Evaluation Metrics.** We evaluate performance from two perspectives:

- **Downstream Task Performance:** The primary metric, measuring the quality of the final answer. We compute token-level F1-Score, Precision, and Recall between the generated answer ($A^*$) and the oracle answer ($A_{oracle}$).
- **Knowledge Chunk Editing Quality:** A secondary metric assessing the precision of the edit itself. We use ROUGE-L (Y., 2004) to measure the similarity between the corrected chunk ($K^*$) and the original perturbed chunk ($K_0$), where a higher score indicates a more minimal, surgical correction.

**Implementation Details.** To ensure a fair comparison, all methods are powered by the same base model. We provide detailed hyperparameters and further implementation specifics in Appendix C.

## 5.2 RESULTS AND ANALYSIS

### 5.2.1 MAIN PERFORMANCE COMPARISON

The aggregated results on both test sets are reported in Table 2. The data clearly indicate that MT-KOC achieves the best performance across all downstream task metrics on both Benchmarks. The consistent performance hierarchy validates the superiority of our multi-step, adaptive search strategy over other structured refinement methods, significantly closing the gap to the Oracle performance.

**Analysis of the Gap to Oracle.** Despite MT-KOC's strong performance, a discernible gap to the Oracle upper bound persists. Our analysis attributes this gap to two primary challenges: 1) the inherent ambiguity of **vague user feedback** ($F$), which could be mitigated by multi-turn clarification dialogues; 2) scenarios where Knowledge Chunks ($K_0$) contain **specialized or lesser-known information**. Both cases compel the framework to fall back upon the base model's own parametric knowledge (Lewis et al., 2020). This indicates that the primary performance bottleneck lies not with the MT-KOC

| Method | ConvCorrect-SQ | | ConvCorrect-NB | |
|---|---|---|---|---|
| | v.s. Perturbed | v.s. Oracle | v.s. Perturbed | v.s. Oracle |
| CRAG | 62.1 | 59.1 | 56.0 | 44.9 |
| RARR | 60.1 | 59.6 | 47.8 | 39.8 |
| Astute | 59.2 | 59.2 | 54.2 | 46.0 |
| **MT-KOC** | **79.7** | **78.4** | **63.1** | **53.4** |

Table 3: ROUGE-L similarity scores on the ConvCorrect-SQ and ConvCorrect-NB test sets.

framework itself, but with the knowledge breadth of the underlying LLM, a limitation that could be addressed by integrating real-time web search capabilities.

### 5.2.2 KNOWLEDGE CHUNK EDITING RESULTS ANALYSIS

To understand how each method edits the Knowledge Chunk, we analyze the ROUGE-L scores, shown in Table 3. Firstly, MT-KOC achieves the highest ROUGE-L score against the Oracle knowledge chunk. This directly measures the **accuracy** of the correction, demonstrating that the final edited chunk ($K^*$) is semantically closest to the ground-truth correct version. Secondly, and equally important, it also achieves the highest score against the original Perturbed chunk. This metric reflects **edit retention**, indicating that our method preserves the maximum amount of correct, unchanged information from the original chunk ($K_0$).

### 5.3 ABLATION STUDY

We conducted an ablation study to validate the contributions of MT-KOC's key components, with results shown in Table 4.

**Ablation on Multi-Path Search.** Removing multi-path exploration by degenerating the process into a linear, greedy search leads to a consistent degradation in performance. This confirms that the search mechanism is crucial for escaping local optima and finding a globally superior correction path.

**Ablation on Shortcut Mechanism.** Disabling the Shortcut mechanism also causes a clear drop in F1-score, which suggests that the model becomes prone to performing superfluous, harmful edits without an effective early termination signal.

| Method | Metric | SQ | NB |
|---|---|---|---|
| **MT-KOC** | F1 | 24.8 | 56.3 |
| | P | 14.9 | 54.1 |
| | R | 73.6 | 58.7 |
| w/o M-P Search | F1 | 24.6 ($\downarrow$ 0.2) | 55.9 ($\downarrow$ 0.4) |
| | P | 14.8 ($\downarrow$ 0.1) | 53.6 ($\downarrow$ 0.5) |
| | R | 72.6 ($\downarrow$ 1.0) | 58.3 ($\downarrow$ 0.4) |
| w/o Shortcut | F1 | 23.3 ($\downarrow$ 1.5) | 55.5 ($\downarrow$ 0.8) |
| | P | 14.0 ($\downarrow$ 0.9) | 53.2 ($\downarrow$ 0.9) |
| | R | 69.3 ($\downarrow$ 4.3) | 57.9 ($\downarrow$ 0.8) |

Table 4: Ablation study of MT-KOC. SQ and NB refer to the ConvCorrect-SQ and ConvCorrect-NB datasets, respectively. P/R denotes Precision/Recall.

## 6 CONCLUSION

In this paper, we introduced the task of KOC to correct errors in RAG knowledge bases in real-time using conversation-based user feedback. To address this task, we developed ConvCorrect, the first comprehensive benchmark that incorporates diverse error types and feedback styles derived from a preliminary user study. Furthermore, we proposed MT-KOC, a multi-step correction framework leveraging a multi-agent system and dynamic action search algorithm to identify and apply optimal sequences of corrective actions. Empirical evaluations on ConvCorrect demonstrate that MT-KOC significantly outperforms existing baselines, achieving higher correction accuracy.

**Ethics Statement** Our research includes a preliminary user study involving human participants. All participants were informed of the study's objectives and procedures before providing their consent to participate. The data collected, which consists of conversation-based feedback, was fully anonymized to protect the privacy of the participants. Each participant received fair compensation for their time, independent of their performance. The datasets used as the foundation for our benchmark, SQuAD and Neural-Bridge, are established public resources, and we have taken additional steps to filter them for any potentially sensitive content.

**Reproducibility Statement** To ensure the reproducibility of our results, we provide detailed descriptions of our benchmark, methodology, and experimental setup. The construction process of our ConvCorrect benchmark, including data sources and processing pipelines, is detailed in Section 3. The architecture and the dynamic action search algorithm of our proposed MT-KOC method are described in Section 4. Full implementation details, including the base model used and hyperparameters, are provided in Appendix C. The source code and benchmark data will be made publicly available upon publication of this work.

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

APPENDIX

## A    SAMPLE WITH EACH ERROR TYPE

**Shared Context for All Examples**

- **User Query (Q):** *Please describe the key identifying features of the Death Cap mushroom.*

- **Oracle Knowledge Chunk ($K_{oracle}$):** *...The Death Cap (Amanita phalloides) is one of the world's most poisonous basidiomycetes. It contains $\alpha$-amanitin, which causes fatal liver and kidney damage. The species typically grows in summer and autumn in deciduous and mixed forests, forming mycorrhizal relationships with broadleaf trees like oaks. Its key identifying features include: a cap that is typically pale grey, yellowish-green, or olive-green, and smooth. The gills are white and free (i.e., not attached directly to the stipe). The most important feature is that the base of its stipe has a distinct, cup-like volva, and the upper part of the stipe usually has an annulus (ring)...*

- **Correct Answer ($A_{oracle}$):** *The key features of the Death Cap are: 1) A cap color of pale grey, yellowish-green, or olive-green; 2) Gills that are white and free; and 3) A cup-like volva structure at the base of the stipe.*

FULLY INCORRECT

- **Erroneous Knowledge Chunk (K):** *...The Death Cap (Amanita phalloides) is one of the world's most poisonous basidiomycetes. It contains $\alpha$-amanitin, which causes fatal liver and kidney damage. The species typically grows in summer and autumn in deciduous and mixed forests. Its key identifying features include: a cap that is typically bright red with white warts. The gills are yellow and decurrent (i.e., running down the stipe). The base of its stipe lacks any special cup-like structure, simply tapering at the base, and the upper part of the stipe usually has an annulus (ring)...*

- **Generated Answer from K (A):** *The key features of the Death Cap include: 1) A cap that is bright red with white warts; 2) Gills that are yellow and decurrent; and 3) No cup-like structure at the base of its stipe.*

PARTIALLY INCORRECT

- **Erroneous Knowledge Chunk (K):** *...The Death Cap (Amanita phalloides) is one of the world's most poisonous basidiomycetes. It contains $\alpha$-amanitin, which causes fatal liver and kidney damage. The species typically grows in summer and autumn in deciduous and mixed forests. Its key identifying features include: a cap that is typically pure white or light brown, and smooth. The gills are white and free (i.e., not attached directly to the stipe). The most important feature is that the base of its stipe has a distinct, cup-like volva, and the upper part of the stipe usually has an annulus (ring)...*

- **Generated Answer from K (A):** *The key features of the Death Cap include: 1) A cap color of pure white or light brown; 2) Gills that are white and free; and 3) A cup-like volva structure at the base of the stipe.*

FULLY MISSING

- **Erroneous Knowledge Chunk (K):** *...The Death Cap (Amanita phalloides) is one of the world's most poisonous basidiomycetes and is responsible for the vast majority of fatal mushroom poisonings worldwide. Its primary toxin is $\alpha$-amanitin, a heat-stable cyclopeptide that cannot be destroyed by cooking. Symptoms of poisoning typically do not appear until 6 to 12 hours after consumption, starting with severe abdominal pain and vomiting. This is followed by a brief period of false recovery, which ultimately leads to catastrophic liver and kidney failure...*

- **Generated Answer from K (A):** *Regarding the Death Cap mushroom, the provided information indicates that it is extremely toxic and causes the majority of fatal mushroom poisonings. Its toxin, $\alpha$-amanitin, is heat-stable, and symptoms of poisoning are delayed, leading to severe liver and kidney failure.*

PARTIALLY MISSING

- **Erroneous Knowledge Chunk (K):** *...The Death Cap (Amanita phalloides) is one of the world's most poisonous basidiomycetes. It contains α-amanitin, which causes fatal liver and kidney damage. The species typically grows in summer and autumn in deciduous and mixed forests. Its key identifying features include: a cap that is typically pale grey, yellowish-green, or olive-green, and smooth. The gills are* light brown *and* decurrent *(i.e., running down the stipe). The upper part of the stipe usually has an annulus (ring)...*

- **Generated Answer from K (A):** *The key features of the Death Cap include: 1) A cap color of pale grey, yellowish-green, or olive-green; and 2) Gills that are white and free.*

MIXED

- **Erroneous Knowledge Chunk (K):** *...The Death Cap (Amanita phalloides) is one of the world's most poisonous basidiomycetes. It contains α-amanitin, which causes fatal liver and kidney damage. The species typically grows in summer and autumn in deciduous and mixed forests. Its key identifying features include: a cap that is typically pale grey, yellowish-green, or olive-green, and smooth. The gills are* light brown *and* decurrent *(i.e., running down the stipe). The upper part of the stipe usually has an annulus (ring)...*

- **Generated Answer from K (A):** *The key features of the Death Cap include: 1) A cap color of pale grey, yellowish-green, or olive-green; and 2) Gills that are light brown and decurrent.*

| ConvCorrect-SQ | | ConvCorrect-NB | |
|---|---|---|---|
| **Statistic** | **Value** | **Statistic** | **Value** |
| **Total Samples** | 124,531 | **Total Samples** | 80,772 |
| **Perturbation Type Distribution** | | **Perturbation Type Distribution** | |
| Replace | 32.8% | Replace | 31.8% |
| Add | 33.5% | Add | 34.8% |
| Delete | 33.7% | Delete | 33.4% |
| **Error Type Distribution** | | **Error Type Distribution** | |
| Fully incorrect | 77.4% | Fully incorrect | 55.5% |
| Partially incorrect | 11.9% | Partially incorrect | 29.7% |
| Fully missing | 0.2% | Fully missing | 0.5% |
| Partially missing | 10.1% | Partially missing | 11.4% |
| Mixed | 0.4% | Mixed | 2.9% |

Table 5: Key statistics of the ConvCorrect Benchmark, detailing the scale and distribution of types across both splits.

## B    PROMPTS FOR MULTI-AGENT SYSTEM

This section provides the detailed prompts used for each agent in the MT-KOC framework. These prompts are presented with maximum fidelity to the implementation code to ensure reproducibility. Each major prompt is presented on a separate page for clarity. Placeholders are denoted with angle brackets (e.g., <query>).

### B.1    PROMPT FOR KNOWLEDGE DISTILLATION AGENT

The agent is prompted to act as a "Factual Extractor." It uses the user's query and the erroneous Knowledge Chunk as a hint to find a minimal, direct, and factual piece of reference information.

```
You are a hyper-focused Factual Extractor.  You will be
given a [Query] and an [Erroneous Knowledge Chunk].  The
[Erroneous Knowledge Chunk] is a previous, flawed attempt to
answer the query.  Your task is to use this erroneous chunk
as a crucial hint to disambiguate the user's true intent.
You must adhere to the following strict rules:

    1. Use Knowledge Chunk for Disambiguation:  Analyze the
       [Erroneous Knowledge Chunk] to understand the specific
       angle or meaning the user is interested in, especially
       for ambiguous queries.

    2. Direct Answer First:  Primary goal is to find a
       direct, minimal, and factual answer to the [Query],
       informed by the hint from the [Erroneous Knowledge
       Chunk].  Extract only the core piece of information
       needed.  For example, if asked for a capital city,
       provide only the city's name.  If asked for a date,
       provide only the date.

    3. Strict Relevance Filter:  Aggressively filter out
       any information that is not essential to answering
       the query.  Ignore historical context, biographical
       details, related trivia, or explanations unless the
       query explicitly asks for them.

    4. Mandatory Generative Fallback:  If, and only if, no
       verifiable information is found, you must generate a
       plausible and concise answer, prefaced with:  "The
       following information is generated for reference only
       and may not be accurate, as no verified data was found
       in the internal knowledge base."

    5. No Empty Responses:  You must always provide a
       response.

    6. Format:  Return the information as plain text.
Examples:
[Query]:  Who was the first person to walk on the moon?
[Erroneous Knowledge Chunk]:  The first person to walk on
the moon was Yuri Gagarin.
[Output]:  Neil Armstrong.
[Query]:  What is Java?
[Erroneous Knowledge Chunk]:  Java is a popular caffeinated
beverage made from roasted coffee beans.
[Output]:  Java is a large island in Indonesia.
```

## B.2 PROMPT FOR ACTION RECOMMENDATION AGENT

The agent is prompted to analyze the user feedback, the potentially erroneous Knowledge Chunk, and the reference information to generate a set of structured correction hypotheses (actions).

```
You are a brilliant AI Recommendation Generator.  Your
task is to analyze user feedback and a potentially flawed
[Knowledge Chunk] to generate plausible correction
hypotheses.  You will be given [Reference Information] to
assist you, but you must use it critically.
Analysis Protocol:
   1. Prioritize User Feedback:  Your primary guide is
      always the user's [Feedback].
         • Specific Feedback:  If the [Feedback] is direct and
           clear (e.g., "The capital is Paris, not Berlin"),
           your main goal is to create a recommendation that
           directly implements this feedback.  The [Reference
           Information] should be considered secondary, used
           only to verify or add minor, consistent details.
           The user's explicit correction takes precedence.
         • Vague Feedback:  If the [Feedback] is vague (e.g.,
           "That's wrong," "It's incomplete"), you must
           critically use the [Reference Information] as your
           main tool to deduce the user's intent.  Compare the
           [Knowledge Chunk] with the [Reference Information]
           to find likely errors or omissions.  Generate
           2-4 distinct hypotheses based on these potential
           discrepancies.  Treat the reference as a strong
           hint, not an absolute truth.
   2. Recommendation Structure:  Each recommendation
      must be a self-contained object with two keys:
      `"description"` and `"action"`.
   3. Action Mapping:  The "action" object must contain:
      `"action_type"` (one of ["REVISE", "ADD", "DELETE"])
      and `"recommendation"`.
   4. Output Format:  Return ONLY a valid JSON array of
      recommendation objects.  If no corrections seem
      necessary, return an empty JSON array `[]`.
Input Format:
[Query]:  The user's original question.
[Previous response]:  The model's previous answer.
[Knowledge Chunk]:  The original Knowledge Chunk that needs
correction.
[Feedback]:  The user's feedback.
[Reference Information]:  Factual information provided for
inspiration and guidance.  Note:  This information is for
reference only and is not guaranteed to be the absolute
truth.
```

### B.3 PROMPTS FOR CORRECTION AGENT

The agent acts as a precise text editor, executing a single action provided by the Action Recommendation Agent. The following pages detail the prompts for each of its modules.

ADD MODULE

```
You are a precise AI text editor.  Your sole task is to
add information to the [Knowledge Chunk] based on a single,
clear [ADD Instruction].
Instructions:
    1. Identify Information:  Read the [ADD Instruction] to
       understand what new information needs to be added.

    2. Execute Insertion:  Insert the new information into
       the most logical and natural position within the
       [Knowledge Chunk].

    3. Strict Adherence:  You MUST add the information
       exactly as provided in the instruction.  Do not alter
       it.

    4. Preserve Everything Else:  All other parts of the
       original Knowledge Chunk must be retained exactly.

    5. Output Plain Text:  Output only the complete, modified
       Knowledge Chunk as plain text.
Example:
[Knowledge Chunk]:  Japan is an island nation in East Asia.
[ADD Instruction]:  Add 'with a population of approximately
125 million' to the description of Japan.
[Output]:  Japan is an island nation in East Asia with a
population of approximately 125 million.
```

REVISE MODULE

You are a precise AI text editor. Your sole task is to
modify the [Knowledge Chunk] based on a single, clear
[REVISE Instruction].
**Instructions**:

1. **Identify Target**: Read the [REVISE Instruction] to
   understand which part of the [Knowledge Chunk] is
   incorrect.

2. **Execute Correction**: Revise the incorrect part
   with the new information provided in the [REVISE
   Instruction].

3. **Strict Adherence**: You **MUST** use the information
   exactly as given in the instruction. Do not add,
   infer, or hallucinate any information not present in
   the instruction.

4. **Preserve Everything Else**: All other parts of the
   Knowledge Chunk must be retained exactly.

5. **Output Plain Text**: Output only the complete, modified
   Knowledge Chunk as plain text.

**Example**:
[Knowledge Chunk]: The capital of France is Florida, a
vibrant city known for its art museums. France is in
Europe.
[REVISE Instruction]: In the Knowledge Chunk, change the
capital of France from 'Florida' to 'Paris'.
[Output]: The capital of France is Paris, a vibrant city
known for its art museums. France is in Europe.

DELETE MODULE

```
You are a precise AI text editor.  Your sole task is to
remove a misleading segment from the [Knowledge Chunk] based
on a single, clear [DELETE Instruction].
Instructions:
    1. Identify Target:  Read the [DELETE Instruction] to
       understand which specific part of the Knowledge Chunk
       is misleading and should be removed.

    2. Execute Deletion:  Remove only the identified
       misleading segment from the [Knowledge Chunk].

    3. Preserve Everything Else:  All other parts of the
       Knowledge Chunk must be retained exactly.

    4. Ensure Coherence:  The remaining text must be
       grammatically correct and coherent.

    5. Output Plain Text:  Output only the complete, modified
       Knowledge Chunk as plain text.
Example:
[Knowledge Chunk]:  The largest planet is Jupiter, which is
a star and has many moons.  It orbits the Sun.
[DELETE Instruction]:  Remove the phrase ', which is a
star,' as it incorrectly describes Jupiter.
[Output]:  The largest planet is Jupiter, which has many
moons.  It orbits the Sun.
```

### B.4 PROMPTS FOR REWARD EVALUATION AGENT

The agent is composed of two modules: one for generating an answer and one for scoring it. The following pages detail the prompts for each module.

ANSWER GENERATION MODULE

```
You are a knowledgeable AI assistant that interacts
with users through natural conversation.  Follow these
guidelines:
    1. Treat the provided Knowledge Chunk as normal and
       complete, and base your responses strictly on it.
       If the Knowledge Chunk lacks relevant details or is
       contradictory, generate a natural and concise answer
       using available information or reasonable inference.

    2. Use accurate terminology and proper names from the
       Knowledge Chunk, paraphrasing naturally for clear and
       conversational responses.

    3. Keep responses concise, addressing the core of the
       question in 1-2 sentences, focusing only on the most
       relevant information.

    4. Present information in a fluent, natural conversation
       style, avoiding phrases like 'the information
       provided' or references to external sources.  For
       missing information, provide a plausible answer based
       on the Knowledge Chunk or inference.

    5. If the Knowledge Chunk contains contradictory
       information, include all relevant details in the
       answer while maintaining natural expression.

    6. Always attempt to answer the question, even if
       the Knowledge Chunk lacks direct information, by
       leveraging related content or reasonable assumptions.
```

SCORING MODULE

You are a strict and objective Answer Evaluator. Your sole purpose is to score a new answer based on its precise alignment with a ground truth. Your evaluation must be rigorous and prioritize factual accuracy over stylistic qualities.

**Evaluation Protocol (Strict Priority Order):**

1. **Determine the Ground Truth:** You must determine the ground truth by following this hierarchy:

   • **Tier 1: Clear Feedback:** If the [User Feedback] is specific and provides a clear correction (e.g., "it should be Paris"), that feedback **is the absolute and sole ground truth**. You MUST ignore the [Reference Information], even if it contains additional or conflicting details.

   • **Tier 2: Vague Feedback with Reference:** If the [User Feedback] is vague (e.g., "it's wrong") **AND** the [Reference Information] is available, then the [Reference Information] **becomes the ground truth**.

   • **Tier 3: Vague Feedback, No Reference (Inference):** If the [User Feedback] is vague **AND** the [Reference Information] is empty or unavailable, you must first use your own general knowledge to determine what a perfect answer to the [Query] would be. This **inferred ideal answer** then becomes your ground truth for the evaluation.

2. **Score based on Ground Truth Alignment (0–10):**

   • **10 (Perfect Match):** The new answer perfectly incorporates the ground truth. All key entities, facts, and numbers from the ground truth are present and correct.

   • **7–9 (High Alignment):** The new answer correctly incorporates the main point of the ground truth but may miss a minor detail.

   • **4–6 (Partial Alignment):** The new answer addresses the ground truth partially (e.g., corrects one error but misses another).

   • **0–3 (Low Alignment):** The new answer attempts to address the ground truth but fails significantly, containing major inaccuracies.

**Scoring Rules:**

   • **Precision is Key:** Focus entirely on the presence and correctness of key information from the ground truth you determined in Step 1.

   • **Output ONLY the numerical score (0–10).**

## C   IMPLEMENTATION DETAILS

To ensure a fair and controlled comparison, all methods evaluated in our experiments, including all baselines and every agent within the MT-KOC framework, are powered by the same underlying large language model.

**Base Model and Environment.**   We use **DeepSeek-R1-Distill-Qwen-32B** as the base model for all experiments. The model is deployed on 8 H100 GPUs using the VLLM framework for efficient inference.

**MT-KOC Hyperparameters.**   For our proposed MT-KOC framework, the number of search epochs is set to 8. The exploration coefficient $E$ in the UCT-based selection metric (Eq. 6) is set to 1.3. The Shortcut mechanism is triggered when an evaluation score $S_i$ from the Reward Evaluation Agent reaches the maximum value of 10.0.

