# OpenReview forum: "RAG Knowledge Online Correction with Conversation-Based User Feedback"
_ICLR.cc/2026/Conference — ICLR 2026 Conference Withdrawn Submission_

### Official Review · Reviewer_tyyx · 2025-10-17

**Soundness:** 2
**Presentation:** 3
**Contribution:** 2
**Rating:** 2
**Confidence:** 5

**Summary:**

This paper proposes a new benchmark ConvCorrect and a new method MT-KOC according to the theory Formative Assessment with Feedback model. For the benchmark, it utilizes two exiting datasets, which named SQuAD and Neural-Bridge, respectively. For MT-KOC, it uses the MCTS-style framework to iterative search the best solution.

**Strengths:**

1、Proposed a new benchmark based on two existing datasets, which is better for the NLP community.

2、Introduced a human-feedback theory for the LLM's error correction.

3、Proposed a new framework based on MCTS.

**Weaknesses:**

1、The related work is not complete, For example Re-Ex[1], PURR[2], ARE[3] and so on.

2、The author mentions the inference time, MT-KOC is more fast than previous work, but MT-KOC uses the MCTS to construct the whole framework, we know that the MCTS is time-consuming. It is unclear what the real Latency is among the different systems.

3、The human feedback $F$ that labeled in benchmark is directly used in MT-KOC?

4、The component of M-P Search looks useless; when removing this module, the performance only decreases 0.1 point on SQ.

5、The retrieval process is unclear. MT-KOC will use the knowledge from LLM and the internet, but the ratio is not reported.

6、The improvement of Aggregated F1-Scores is relative small.

If all concerns be solved, I will rise my score.

[1]Kim J, Lee J, Chang Y H, et al. Re-Ex: Revising after Explanation reduces the Factual Errors in LLM Responses[C]//ICLR 2024 Workshop on Reliable and Responsible Foundation Models.

[2]Chen A, Pasupat P, Singh S, et al. Purr: Efficiently editing language model hallucinations by denoising language model corruptions[J]. arXiv preprint arXiv:2305.14908, 2023.

[3]Yan Z, Wang J, Chen J, et al. Atomic fact decomposition helps attributed question answering[J]. IEEE Transactions on Knowledge and Data Engineering, 2025.

**Questions:**

1、Can you provide the Latency?

2、Can you provide the ratio in retrieval process？

---

### Official Review · Reviewer_VmZu · 2025-10-22

**Soundness:** 3
**Presentation:** 4
**Contribution:** 4
**Rating:** 6
**Confidence:** 4

**Summary:**

This paper introduces the task of Knowledge Online Correction (KOC), which aims to correct erroneous information in retrieval-augmented generation (RAG) systems in real time using user feedback. Traditional RAG pipelines assume accurate retrieval sources, but when knowledge bases contain errors, manual correction is slow and leads to persistent misinformation.

To address this, the authors propose leveraging conversation-based feedback, where users indicate or correct factual mistakes during interaction. They conduct a user study identifying five error types and seven feedback styles, and use these insights to build ConvCorrect, a large-scale benchmark based on perturbed versions of SQuAD and Neural-Bridge datasets.

The proposed method, MT-KOC, is a multi-agent framework that performs iterative correction through a dynamic action search mechanism. It employs four agents (knowledge distillation, action recommendation, correction, and evaluation) to identify and apply optimal edit sequences (ADD, DELETE, REVISE).

Experiments on ConvCorrect show that MT-KOC outperforms all baselines (CRAG, RARR, Astute), and produces more accurate and minimal edits. The paper’s main contributions are: (1) defining the KOC task, (2) releasing the ConvCorrect benchmark, and (3) proposing MT-KOC, an effective framework for feedback-driven online knowledge correction.

**Strengths:**

The paper is well-motivated and tackles a timely and increasingly relevant problem: how to correct non-parametric knowledge in RAG systems when external sources contain inaccuracies. As LLMs rely more heavily on retrieval-based augmentation, the need for mechanisms that can automatically detect and repair knowledge errors becomes both practically significant and underexplored. The proposed task of Knowledge Online Correction (KOC), therefore, fills an important research gap and represents a clear step toward more reliable and adaptive RAG systems.

The proposed benchmark ConvCorrect is comprehensively constructed, grounded in a user study that identifies diverse feedback styles and error types, and supported by systematic perturbation procedures and quality verification. Its large scale and diversity make it a strong foundation for future research in this area.

In terms of methodology and quality, the proposed MT-KOC framework is well-structured and conceptually clear. The modular, multi-agent design, decomposing the correction process into retrieval, recommendation, editing, and evaluation, reflects careful system design and facilitates interpretability. The dynamic action search mechanism adds sophistication by enabling adaptive exploration of edit sequences, rather than relying on single-step or static corrections.

The experimental section is solid, covering multiple baselines and including key ablation studies that isolate the effects of core components such as the multi-path search and shortcut mechanism. Results are consistent across datasets and clearly support the paper’s claims.

Finally, the paper is clearly written and well-organized, with a logical narrative from motivation to benchmark construction, method design, and experiments.

**Weaknesses:**

A first concern lies in the user study design, which forms the empirical foundation for the benchmark. All 12 participants are computer science graduate students with prior experience using RAG systems. This creates a strong sampling bias, as such participants likely provide structured, technically precise feedback that does not reflect how ordinary users express confusion or corrections. One of the key challenges in leveraging conversation-based feedback lies precisely in its noisiness and ambiguity. Therefore, basing the benchmark taxonomy and feedback styles solely on technically proficient users risks overestimating the method’s robustness in real-world applications. To strengthen the validity of the ConvCorrect benchmark, future iterations should include a broader and more heterogeneous participant pool and analyze how feedback quality and phrasing vary across user types.

Second, the feedback generation process remains underspecified. Although the paper states that LLMs were used to simulate user feedback in various styles, it does not clarify which model was used or how prompts were constructed to ensure natural, human-like responses. Given that much of the benchmark’s realism depends on the diversity and authenticity of this generated feedback, the absence of such details weakens reproducibility and makes it difficult to assess whether the benchmark truly captures realistic user behavior. Including concrete examples of generated feedback, prompt templates, and a justification for the chosen model would substantially improve transparency.

Third, the terminology around “knowledge distillation” in Section 4.2 is misleading. The described Knowledge Distillation Agent appears to retrieve factual information for reference, yet the term “knowledge distillation” in machine learning traditionally refers to training a smaller model to imitate a larger one [1]. The authors should either adopt a more accurate name or clearly explain in what sense this component performs distillation, ideally citing canonical works on distillation to avoid confusion.

Finally, the experimental setup lacks important implementation details. Section 5.1 does not specify the underlying LLM architecture used for MT-KOC and for the baselines (at least this information should be moved from the Appendix to the main paper). Moreover, all results appear to rely on a single base model, which limits the conclusions that can be drawn about the framework’s generality. Evaluating the approach across multiple LLMs of different scales or families would provide stronger evidence that the proposed method generalizes beyond one specific model setup.

Overall, while the conceptual framing and benchmark design are promising, the paper would benefit from greater methodological transparency, broader user diversity, and clarified terminology to fully substantiate its contributions.

**Questions:**

1) The user study forms the foundation of the ConvCorrect benchmark, yet all participants were computer science graduate students familiar with RAG systems. How do the authors justify that the resulting taxonomy of error types and feedback styles generalizes to real-world users, who may provide shorter, noisier, or less structured feedback? Have the authors considered conducting follow-up studies with a more diverse participant pool to validate these findings?
2) The paper does not specify how conversation-based feedback was simulated once the user study results were used to guide feedback generation. Which LLM was used for this step, and how were prompts designed to ensure that the generated feedback is natural and consistent with human linguistic variability? Clarifying this would improve reproducibility and help assess the realism of the ConvCorrect dataset.
3) The experiments seem to rely on a single LLM backend. How sensitive is the MT-KOC framework to the choice of base model? Would the method perform similarly if built upon a smaller or instruction-tuned model, or does it rely on strong reasoning capabilities specific to a particular model family?
4) Since the approach depends heavily on user feedback as a correction signal, how robust is MT-KOC to cases where the feedback is vague, misleading, or even intentionally harmful? Have the authors considered mechanisms such as feedback verification or confidence estimation to mitigate erroneous corrections?
5) The proposed task introduces an interesting security dimension: if user feedback can alter knowledge chunks, it may also be exploited to inject misinformation. How do the authors envision safeguarding against such adversarial or coordinated manipulation in real-world systems? Are there mechanisms in the framework to detect or filter suspicious feedback?

---

### Official Review · Reviewer_AQiP · 2025-10-29

**Soundness:** 3
**Presentation:** 3
**Contribution:** 3
**Rating:** 4
**Confidence:** 3

**Summary:**

This paper introduces a critical new task called Knowledge Online Correction (KOC), aiming to address the degradation of performance and poor user experience in Retrieval-Augmented Generation (RAG) systems caused by errors in the knowledge base. The authors argue that the traditional manual correction process is often time-consuming, taking hours or even days. To enable evaluation of this task, the authors developed the first comprehensive benchmark, ConvCorrect, derived from a preliminary user study that identified five major error types and seven conversational feedback styles. To tackle KOC, the paper proposes MT-KOC (MulTi-step KOC), a multi-agent system utilizing a dynamic action search algorithm to identify and apply the optimal sequence of corrective actions in real time. Empirical results show that MT-KOC significantly outperforms existing baselines, achieving higher correction accuracy

**Strengths:**

1. The KOC task is highly relevant to real-world deployment of RAG systems. By focusing on real-time correction based on user feedback, the work directly tackles the poor user experience caused by knowledge staleness and manual intervention. The comparison illustrated in Figure 1 effectively highlights the value of this new paradigm.

2. The methodology for creating ConvCorrect, starting with a preliminary user study to derive error types and feedback styles, is excellent. The finding of a complex "one-to-many" mapping between error types and feedback styles is crucial, as it validates the benchmark's realism and prevents simplistic generation strategies.

3. The MT-KOC framework successfully adapts a search-based multi-agent system (inspired by MCTS/UCT) to a corrective task. Decomposing the correction into discrete, traceable actions (ADD, DELETE, REVISE) and using a reward-driven search strategy is an elegant solution to multi-step knowledge editing.

**Weaknesses:**

1. The quality of the benchmark fundamentally relies on the LLM acting as a User Simulator generating the conversation-based feedback ($F$). The paper does not provide details or an evaluation of the fidelity of this LLM-based User Simulator. How was the quality, diversity, and realism of the simulated feedback validated beyond the initial human-verified classification of error types? A human evaluation or case study comparing the LLM-generated feedback to the user study's collected feedback is necessary to validate the benchmark's quality.

2. The experimental setup is confined to a single base model, DeepSeek-R1-Distill-Qwen-32B, which powers all agents and baselines.

3.  If a user provides clear but factually incorrect feedback (e.g., correcting "Paris" to "London" when the true capital is "Tokyo"), MT-KOC is designed to accept this error. Could the authors discuss this vulnerability, and if a conflict detection mechanism (e.g., scoring $F$ against $I$ before setting $F$ as the absolute truth) could be beneficial, even if it adds complexity?

4. The modification of the knowledge chunk K relies on the retrieved documents, which may cause the model to incorrectly update the knowledge. How to deal with this problem?

**Questions:**

See Weakness.

---

### Official Review · Reviewer_q7HB · 2025-10-31

**Soundness:** 3
**Presentation:** 3
**Contribution:** 2
**Rating:** 4
**Confidence:** 3

**Summary:**

The paper introduces **Knowledge Online Correction (KOC)**; a new task for **real-time correction of factual errors in RAG systems** using **conversation-based user feedback**. It proposes **MT-KOC**, a **multi-agent framework** that dynamically identifies and applies corrective actions (ADD, DELETE, REVISE) through a **reward-guided search** process. To evaluate this task, the authors build **ConvCorrect**, the first benchmark combining **human-derived error types and feedback styles** across over 200k samples from SQuAD and Neural-Bridge datasets. Experiments show that MT-KOC **outperforms existing correction methods (CRAG, RARR, Astute)** by an average **+5.88% F1**, with strong precision/recall and minimal-edit quality.

**Key Contributions**
1. Definition of the **KOC task** for online, feedback-driven knowledge correction.
2. Creation of the **ConvCorrect benchmark** with diverse, realistic feedback–error mappings.
3. Proposal of **MT-KOC**, a **multi-step, dynamic-action correction model** achieving state-of-the-art results.

**Strengths:**

* Introduces a novel task definition; Knowledge Online Correction (KOC); enabling real-time, conversation-based correction of retrieved knowledge via user feedback. Prior works typically focus on offline knowledge base edits or batch-updates (for example, STACKFEED [A] uses expert feedback to edit a KB document by document offline).
* Provides a comprehensive benchmark: ConvCorrect. Prior works such as SKR [B] focus on filtering/rewriting retrieved knowledge but don’t provide a dedicated conversational feedback-to-edit benchmark.
* Proposes MT-KOC, a multi-agent framework combining retrieval, action recommendation, correction and reward evaluation in online feedback loops. Whereas STACKFEED uses a multi-actor RL setup for document-level edits and SKR uses rewriting/filtering modules, few prior works handle interactive editing triggered by user feedback during conversation.
* The paper reports meaningful gains over baselines in the online correction setting. By comparison,SynCheck [C] monitors faithfulness and triggers interventions, but doesn’t itself perform edits of the retrieval knowledge base based on user feedback.

```
[A] STACKFEED: Structured Textual Actor-Critic Knowledge Base Editing with FeedBack, ArXiv 2024
[B] Supportiveness-based Knowledge Rewriting for Retrieval-augmented Language Modeling, ArXiv 2024
[C] Synchronous Faithfulness Monitoring for Trustworthy Retrieval-Augmented Generation, EMNLP 2024
```

**Weaknesses:**

* Although the task is defined as real-time correction via conversation-based user feedback, the experiments appear to focus on pre-constructed feedback types rather than real users in the loop. This raises questions about how well the system performs under noisy, unpredictable user feedback.
* The benchmark (ConvCorrect) is a strength, but if it is constructed from specific datasets (e.g., SQuAD, Neural-Bridge) there may be limited domain diversity, retrieval complexity, and multi-hop knowledge errors. Many prior works (e.g., RAE [A]) explore broader domains and more complex edits.
* The multi-agent, search-based architecture (ADD, DELETE, REVISE via search) may incur substantial computational overhead, which could undermine its suitability for real-time online deployment. Prior works such as CRAG [B] propose lighter-weight evaluations of retrieval quality.
* Real-time corrections may interact with future retrievals and generation behaviour (e.g., a correction might change how future queries are grounded). The paper appears to focus on isolated correction episodes rather than tracking how corrections propagate and whether they introduce inconsistencies downstream.
* While the taxonomy of feedback styles is commendable, it's unclear whether the feedback encompasses the full realistic spectrum of user behaviour (e.g., vague comments, contradictory feedback, no feedback).
* The system is pitched for real-time correction, but the paper may not sufficiently address realistic deployment constraints (e.g., retrieval index updates, user privacy, latency in real conversation, scaling to large corpora).
```
[A] Retrieval-enhanced Knowledge Editing in Language Models for Multi-Hop Question Answering, CIKM 2024
[B] Corrective Retrieval Augmented Generation, ArXiv 2024
```

**Questions:**

1. Is it possible to include a user-study with actual interactive sessions (rather than simulated feedback) to evaluate robustness to varied feedback phrasing, delays, erroneous feedback, or conflicting feedback. Report metrics such as correction latency, user satisfaction, and system resilience to sub-optimal feedback?
2. Is it possible to expand evaluation to multiple domains (e.g., long-tail specialized domains, multi-hop reasoned knowledge) and retrieval corpora (documents, knowledge graphs, dynamic web sources)? This will demonstrate generality of the proposed method.
3. Is it possible to report detailed latency and resource usage (e.g., number of agent steps, search branching, memory/traversal costs) in a realistic deployment scenario? It might be then worthwhile to consider optimizing for runtime (e.g., early stopping heuristics, agent budget limits, selective invocation only when feedback indicates high risk).
4. Is it possible to include experiments tracking how corrections affect subsequent retrieval/generation rounds (e.g., over multiple turns or sessions)? It woud be good to measure whether corrected knowledge persists, whether retrieval biases shift, and whether unintended side-effects occur (e.g., over-correction, drift from oracle knowledge).
5. Is it possible to augment ConvCorrect or supplement experiments with "imperfect feedback" scenarios (typos, ambiguous, sarcastic, irrelevant feedback) and evaluate how the system handles these? It would be good to report failure modes and design fallback strategies (e.g., ask clarifying questions, revert changes).
6. I wonder how easy it is to simulate a "live" retrieval system (e.g., dynamic knowledge streams, index refresh, multiple users) and report how the system behaves under load, with full pipeline latency, and in presence of changing knowledge (e.g., emerging facts). It might be worthwhile to discuss practical deployment challenges and how the system might be scaled.

---

### Note · Authors · 2025-11-22

I have read and agree with the venue's withdrawal policy on behalf of myself and my co-authors.